# Lifespan variation among people with a given disease or condition

Yan Zheng[1], Iñaki Permanyer[2,3], Vladimir Canudas-Romo[4], José Manuel Aburto[5,6,7], Andrea Nigri[8], Oleguer Plana-Ripoll[1] *

1 Department of Clinical Epidemiology, Aarhus University and Aarhus University Hospital, Aarhus, Denmark, 2 Centre d'Estudis Demogràfics, Cerdanyola del Vallès, Barcelona, Spain, 3 ICREA, Passeig Lluís Companys 23, Barcelona, Spain, 4 School of Demography, ANU College of Arts & Social Sciences, Australian National University, Canberra, Australia, 5 Department of Population Health, London School of Hygiene and Tropical Medicine, London, United Kingdom, 6 Leverhulme Centre for Demographic Science, Department of Sociology and Nuffield College, University of Oxford, Oxford, United Kingdom, 7 Interdisciplinary Centre on Population Dynamics, University of Southern Denmark, Odense, Denmark, 8 Department of Economics, Management and Territory, University of Foggia, Foggia, Italy

* opr@clin.au.dk

## Abstract

In addition to fundamental mortality metrics such as mortality rates and mortality rate ratios, life expectancy is also commonly used to investigate excess mortality among a group of individuals diagnosed with specific diseases or conditions. However, as an average measure, life expectancy ignores the heterogeneity in lifespan. Interestingly, the variation in lifespan–a measure commonly used in the field of demography–has not been estimated for people with a specific condition. Based on recent advances in methodology in research within epidemiology and demography, we discuss two metrics, namely, the average life disparity and average lifetable entropy after diagnosis, which estimate the variation in lifespan for time-varying conditions in both absolute and relative aspects. These metrics are further decomposed into early and late components, separated by their threshold ages. We use mortality data for women with mental disorders from Danish registers to design a population-based study and measure such metrics. Compared with women from the general population, women with a mental disorder had a shorter average remaining life expectancy after diagnosis (37.6 years vs. 44.9 years). In addition, women with mental disorders also experienced a larger average lifespan variation, illustrated by larger average life disparity (9.5 years vs 9.1 years) and larger average lifetable entropy (0.33 vs 0.27). More specifically, we found that women with a mental disorder had a larger early average life disparity but a smaller late average life disparity. Unlike the average life disparity, both early and late average lifetable entropy were higher for women with mental disorders compared to the general population. In conclusion, the metric proposed in our study complements the current research focusing merely on life expectancy and further provides a new perspective into the assessment of people's health associated with time-varying conditions.

**Data Availability Statement:** Data presented in this study were obtained from Danish registries. Owing to data protection rules, we are not allowed to share individual-level data. Other researchers who fulfill the requirements set by the data

providers could gain access to the data through Statistics Denmark (www.dst.dk) and/or the Danish Health Data Authority (www. sundhedsdatastyrelsen.dk).

**Funding:** This work was supported by a Lundbeck Foundation Fellowship to Oleguer Plana-Ripoll (R345-2020-1588). Oleguer Plana-Ripoll has also received funding from Independent Research Fund Denmark (grants 1030-00085B and 2066-00009B). The funders of the study had no role in developing the methodology, study design, data collection, data analysis, data interpretation, or writing of the report. There was no additional external funding received for this study.

**Competing interests:** The authors have declared that no competing interests exist.

## Introduction

In the field of epidemiology, mortality metrics are fundamental for decision-making, resource distribution in the healthcare sector, and designation of policies related to healthcare planning and social development. Over the past decades, metrics such as mortality rates and mortality rate ratios have been commonly reported in epidemiological studies to evaluate the health outcomes of people diagnosed with specific conditions [1–3]. However, an alternative and more intuitive mortality metric is life expectancy, which can be interpreted as the average number of years a synthetic cohort of newborns (or individuals at a specific age) are expected to live if they were to experience all age-specific mortality rates in that year [4]. As a summary measure for population health, life expectancy has been widely discussed in demographic research and also used by epidemiologists to investigate the excess mortality among individuals diagnosed with specific health conditions for many years [5–8]. The gap in life expectancy between those diagnosed and the general population is an important indicator of the effectiveness of health policies and services targeted at disadvantaged populations [9], which provides implications for the potential improvement in healthcare in the future.

A correct interpretation of the measure of life expectancy requires the assumption that population characteristics are stable over time. Thus, it is not straightforward to calculate this measure for individuals suffering from a specific disease given the time-varying nature of many health conditions. Several studies addressed this concern by estimating life expectancy at a fixed age of onset when exploring particular diseases, e.g., age 15 for mental disorders and age 20 for type 1 diabetes [5, 10]. However, this is equivalent to assuming that everyone with these disorders has onset at that fixed age. Recent studies overcame this limitation by incorporating the observed age-of-onset distribution through averages of age-specific estimates [4, 11, 12]. This metric, known as average remaining life expectancy after diagnosis, has provided more valid results of the reduction in life expectancy associated with specific disorders than that from previous studies [13–17].

Although life expectancy and average life expectancy after diagnosis are informative metrics that policymakers can interpret directly, as average measures, they conceal the heterogeneity in lifespan within populations. Alongside life expectancy, the variation in individual mortality trajectories, which is captured by lifespan variation, has also been widely acknowledged in demographic studies [18–20]. Lifespan variation matters as different populations may have different underlying lifespan distributions, despite their similar life expectancies [21]. Variation in lifespan reveals one of the most fundamental inequalities in human populations [19], and has been proven to have substantial implications at both individual and societal levels, for instance, affecting individual and public investments in education and training [19, 22]. However, despite being a common metric used in recent studies to investigate social inequalities in the length of life [23, 24], lifespan variation has not been estimated for individuals experiencing a particular disease or condition. The lack of such studies is likely to be explained by the challenges of incorporating the age-of-onset into these estimates. Rather than merely focusing on average measures, identifying whether individuals experiencing a particular disease face greater variation in the eventual time of death than the general population would provide a broader understanding of the impact of those diseases or conditions on mortality.

The objective of this study is to propose an informative mortality-related metric that helps evaluate the variation in lifespan for those diagnosed with a given disease or condition. The average lifespan variation after disease diagnosis allows for filling a gap that may not be revealed by current studies focusing on life expectancy only and provides a new perspective for exploring health inequalities related to specific diagnoses. Additionally, the combination of

demographic and epidemiological perspectives provides a novel insight and a more accurate evaluation of the impact of disorders or health conditions on individuals' lifespan distribution.

## Materials and methods

### Empirical data

We describe the method by estimating the average lifespan variation after disease diagnosis among women with mental disorders in Denmark as an illustration. Information on births, immigrations, emigrations, and mortality for this population was obtained from the Danish Civil Registration System [25, 26], and data on mental disorders were collected from the Danish Psychiatric Central Research Register [27]. The two registers were linked on an individual level using a unique personal identification number, which was assigned to all legal residents at birth or immigration to Denmark [26]. We designed a population-based study including all 358,267 women living in Denmark at some point between January 1, 1995 and December 31, 2018, who received a first-time diagnosis of a mental disorder (ICD-10 codes F00-F99) during that period. They were followed up from the date of diagnosis until death, emigration from Denmark or December 31st, 2018, whichever came first. The average age of disease diagnosis was 37.9 years (standard deviation 22.1 years). At the end of follow-up, 61,820 women had died, and the average age at death was 78.4 years (standard deviation 14.3 years). Life tables for women with mental disorders in this cohort are available in Table 1 (selected ages) and S1 Table (all ages). We compared results in the cohort of women diagnosed with mental disorders with estimates from the general female population, which could be calculated based on publicly available mortality information (www.statbank.dk). For this study, we used the life table based on mortality data for the years 2006–2007 (S2 Table), which is the middle point of the study follow-up period. R codes for the analysis are available in S1 File. This study was registered with the Danish Data Protection Agency at Aarhus University (record no. 2016-051-000001-2587) and was approved by Statistics Denmark and the Danish Health Data Authority. In Denmark, no ethical approval or informed consent is required for register-based research. All data was anonymized and specific individuals could not be identified.

### Metrics

**Average life expectancy after disease diagnosis.** The linkage between certain diseases/conditions and life expectancy has been discussed by many studies to examine the excess mortality among those diagnosed with a particular condition. Evidence from those studies highlighted the gap in life expectancy between diagnosed individuals and the comparison group (people without these diseases/conditions or the general population). For instance, compared with the general population, women and men with any type of mental disorder had 12 and 16 years of reduced life expectancy [7], and those with type 1 diabetes reported an estimated reduction of 10–12 years [5]. These studies, however, estimated remaining life expectancy at a fixed age (15 years for mental disorders [7, 10] and 20 years for type 1 diabetes [5]), which is equivalent to assuming that all cases were diagnosed at that age. Such limitation was overcome recently with the introduction of a new method [11], which suggests estimating the life expectancy at each possible age of diagnosis, and then averaging all estimates. The average life expectancy after diagnosis has gained more attention [12–17], and it provides a new perspective to evaluate the premature mortality associated with a particular disease or health condition. Mathematically, average life expectancy after diagnosis can be expressed as

$$\bar{e} = \frac{\sum_{i=1}^{n} e(x_i)}{n},$$ (1)

**Table 1. Life table for women diagnosed with mental disorders based on observed mortality rates.**

| age | lx | dx | Lx | Tx | ex | ed |
|---|---|---|---|---|---|---|
| 15 | 99,941 | 26 | 99,928 | 5,776,393 | 57.8 | 10.9 |
| 16 | 99,915 | 17 | 99,906 | 5,676,466 | 56.8 | 10.9 |
| 17 | 99,898 | 27 | 99,884 | 5,576,559 | 55.8 | 10.9 |
| 18 | 99,871 | 41 | 99,850 | 5,476,675 | 54.8 | 10.9 |
| 19 | 99,829 | 48 | 99,805 | 5,376,825 | 53.9 | 10.8 |
| 20 | 99,781 | 59 | 99,752 | 5,277,020 | 52.9 | 10.8 |
| 21 | 99,723 | 60 | 99,693 | 5,177,268 | 51.9 | 10.8 |
| 22 | 99,663 | 62 | 99,632 | 5,077,575 | 50.9 | 10.8 |
| 23 | 99,601 | 66 | 99,568 | 4,977,943 | 50 | 10.8 |
| 24 | 99,536 | 55 | 99,508 | 4,878,375 | 49 | 10.7 |
| 25 | 99,480 | 46 | 99,457 | 4,778,867 | 48 | 10.7 |
| 26 | 99,434 | 65 | 99,402 | 4,679,410 | 47.1 | 10.7 |
| 27 | 99,370 | 56 | 99,342 | 4,580,008 | 46.1 | 10.7 |
| 28 | 99,314 | 83 | 99,272 | 4,480,666 | 45.1 | 10.6 |
| 29 | 99,231 | 88 | 99,186 | 4,381,394 | 44.2 | 10.6 |
| 30 | 99,142 | 85 | 99,100 | 4,282,207 | 43.2 | 10.6 |
| ... | ... | ... | ... | ... | ... | ... |
| 90 | 7,133 | 1,555 | 6,356 | 22,641 | 3.2 | 2.5 |
| 91 | 5,578 | 1,382 | 4,887 | 16,285 | 2.9 | 2.4 |
| 92 | 4,196 | 1,084 | 3,654 | 11,398 | 2.7 | 2.2 |
| 93 | 3,112 | 912 | 2,656 | 7,744 | 2.5 | 2.1 |
| 94 | 2,200 | 702 | 1,849 | 5,088 | 2.3 | 2 |
| 95 | 1,498 | 504 | 1,246 | 3,239 | 2.2 | 1.9 |
| 96 | 994 | 357 | 815 | 1,993 | 2 | 1.8 |
| 97 | 637 | 241 | 516 | 1,178 | 1.8 | 1.7 |
| 98 | 395 | 172 | 309 | 662 | 1.7 | 1.6 |
| 99+ | 223 | 223 | 353 | 353 | 1.6 | 1.6 |

Source: Authors' calculations based on mortality data from Danish Registers

Note. l_x = survivors to age x, d_x = deaths at age x, L_x = person-years lived at age x, T_x = person-years lived above age x, e_x = life expectancy at age x, ed = life disparity at age x. The life table was calculated based on mortality data from 0 to 99+ years from a cohort of women diagnosed with mental disorders. The cohort was followed between January 1, 1995 and December 31, 2018, covering all individuals living in Denmark. Results are not presented for ages below 15 years due to a low number of individuals. The entire life table is available in the supplement.

where $x_i$ represents the age of onset for individual $i$ and $e(x_i)$ is the remaining life expectancy at that specific onset age, which can be calculated based on life table parameters (as shown in S1 File). Alternatively, if one assumes that age of onset can only take integer values, the average life expectancy after disease diagnosis can also be expressed as a weighted average

$$\bar{e} = \frac{\sum_{x=m}^{M} w(x) \cdot e(x)}{n}, \qquad (2)$$

where $m$ and $M$ represent the minimum and maximum possible ages of onset for the given disease and $w(x)$ represents the number of cases at each age. The sum of the number of cases at each age equals the total number of cases $n = \sum_{x=m}^{M} w(x)$.

With this method, it is possible to estimate the average life expectancy after diagnosis for those with a specific disease or condition, and an estimate can also be obtained for an age-

matched comparison group using the same weights. It is important to use the same weights for the two groups so that the comparison is between groups of the same ages; thus, this average estimate can be conceptualized as a type of standardization. These formulas are based on mortality rates in a given period, but we are ignoring the time component in this study as it would unnecessarily complicate the descriptions. In future studies, it is important to consider potential time trends in such metrics. In our study, for women with mental disorders in Denmark, the average remaining life expectancy after disease diagnosis was 37.6 years. For the general population, the estimate taking into account the observed age-of-onset of those with a disease can be interpreted as the average life expectancy for those of the same age. In this case, the average life expectancy in the general population was 44.9 years, 7.3 years higher than for women with mental disorders.

**Average life disparity after disease diagnosis.**   Lifespan variation can be measured by multiple metrics. Life disparity, one of the absolute indicators, has been commonly used in recent years within the demography area to estimate the variation in lifespan [18, 21] due to its mathematical properties and important public health interpretation [18]. Vaupel and colleagues [20] presented and interpreted this measure as the average remaining life expectancy at the ages when death occurs, or the average life years lost when death occurs. Similar to the approach of estimating average life expectancy, we propose the average life disparity after disease diagnosis, which is defined as the average life disparity at each possible age of onset, weighted by the number of incident cases at that age:

$$\overline{e^{\dagger}} = \frac{\sum_{x=m}^{M} w(x) \cdot e^{\dagger}(x)}{n} , \tag{3}$$

where $e^{\dagger}(x)$ represents the life disparity for each specific age of onset $x$, and can be calculated based on parameters in life tables, and $w(x)$ is the number of cases at the corresponding ages. For women with mental disorders, the average life disparity after disease diagnosis was 9.5 years, 0.4 years larger than women from the general population (9.1 years). Thus, women with a diagnosis of a mental disorder face greater average lifespan variation and thus larger uncertainty at the time of death than those from the general population.

**Difference in early and late average life disparity.**   Unlike life expectancy, which increases when reducing mortality at any age, life disparity declines when delaying premature deaths, while it increases when avoiding deaths at older ages [19, 20]. This is separated by a threshold age, defined as $a^{\dagger}$, after which reducing deaths would start increasing disparity instead of reducing it [28]. Therefore, the overall life disparity can be decomposed into two components regarding the impact of mortality reduction, which was described as the compression component and the expansion component [29]. If individuals suffering from a specific disease have systematically higher mortality rates than the general population (as is the case for those with mental disorders [12, 15]), then life disparity might be larger for those with the disease before a particular threshold age, but smaller than the general population after that age. For this reason, we propose to further separate the average life disparity after disease diagnosis into two components (separated by a threshold age $a^{\dagger}$):

$$\overline{e_0^{\dagger}} = \frac{\sum_{x=m}^{a^{\dagger}} w(x) \cdot e^{\dagger}(x)}{n_0} = \frac{\sum_{x=m}^{a^{\dagger}} w(x) \cdot e^{\dagger}(x)}{\sum_{x=m}^{a^{\dagger}} w(x)}, \tag{4}$$

and

$$\overline{e_1^\dagger} = \frac{\sum_{x=a^\dagger}^{M} w(x) \cdot e^\dagger(x)}{n_1} = \frac{\sum_{x=a^\dagger}^{M} w(x) \cdot e^\dagger(x)}{\sum_{x=a^\dagger}^{M} w(x)}, \qquad (5)$$

where $\overline{e_0^\dagger}$ and $\overline{e_1^\dagger}$ are the average early life disparity and late life disparity after diseases diagnosis, respectively, and could be used to measure the lifespan variation before and after the threshold age $a^\dagger$ for people diagnosed with a given disease or condition. $w(x)$ is the number of diagnosed cases at each age of onset and $n_0 = \sum_{x=m}^{a^\dagger} w(x)$ and $n_1 = \sum_{x=a^\dagger}^{M} w(x)$ represent the number of people diagnosed before and after the threshold age $a^\dagger$, with the sum equal to the total number of diagnosed cases $n$. The threshold age $a^\dagger$ is defined using the mortality rates for those with a condition (the R code for this calculation is available in S1 File).

**Average lifetable entropy after diagnosis.** As a relative indicator of lifespan variation, lifetable entropy is also a useful tool for comparing different patterns of age-at-death distributions [30, 31]. For a specific age $x$, lifetable entropy can be mathematically defined as

$$\overline{H_x} = \frac{e^\dagger(x)}{e(x)}, \qquad (6)$$

Similar to the average life disparity, women diagnosed with mental disorders also had greater average lifetable entropy (0.33) than those in the general population (0.27), indicating their larger inequality in individual lifespan from a relative perspective. Since there is a unique threshold age separating reductions and increases in lifespan variation as a result of age-specific mortality improvements for lifetable entropy [32], we further separated it into two components. After considering the threshold age $h^\dagger$, we would have the early and late average lifetable entropy after diagnosis, respectively, as described below

$$\overline{H_0} = \frac{\sum_{x=m}^{h^\dagger} w(x) \cdot H(x)}{n_0} = \frac{\sum_{x=m}^{h^\dagger} w(x) \cdot H(x)}{\sum_{x=m}^{h^\dagger} w(x)}, \qquad (7)$$

and

$$\overline{H_1} = \frac{\sum_{x=h^\dagger}^{M} w(x) \cdot H(x)}{n_1} = \frac{\sum_{x=h^\dagger}^{M} w(x) \cdot H(x)}{\sum_{x=h^\dagger}^{M} w(x)} \qquad (8)$$

where $\overline{H_0}$ and $\overline{H_1}$ are the average early lifetable entropy and late lifetable entropy after diseases diagnosis, respectively, and could be used to measure the relative variation in lifespan before and after the threshold age $h^\dagger$ for people diagnosed with a given disease/condition. In these formulas, $w(x)$ is the number of cases at each age of onset, and $n_0 = \sum_{x=m}^{h^\dagger} w(x)$ and $n_1 = \sum_{x=h^\dagger}^{M} w(x)$ represent the number of people diagnosed before and after the threshold age $h^\dagger$, with the sum equal to the total number of cases $n$ for the disease. The threshold age $h^\dagger$ is defined using the mortality rates for those with a condition.

## Results

Fig 1 presents the age-specific remaining life expectancy and life disparity for women diagnosed with a mental disorder and those from the general female population in Denmark.

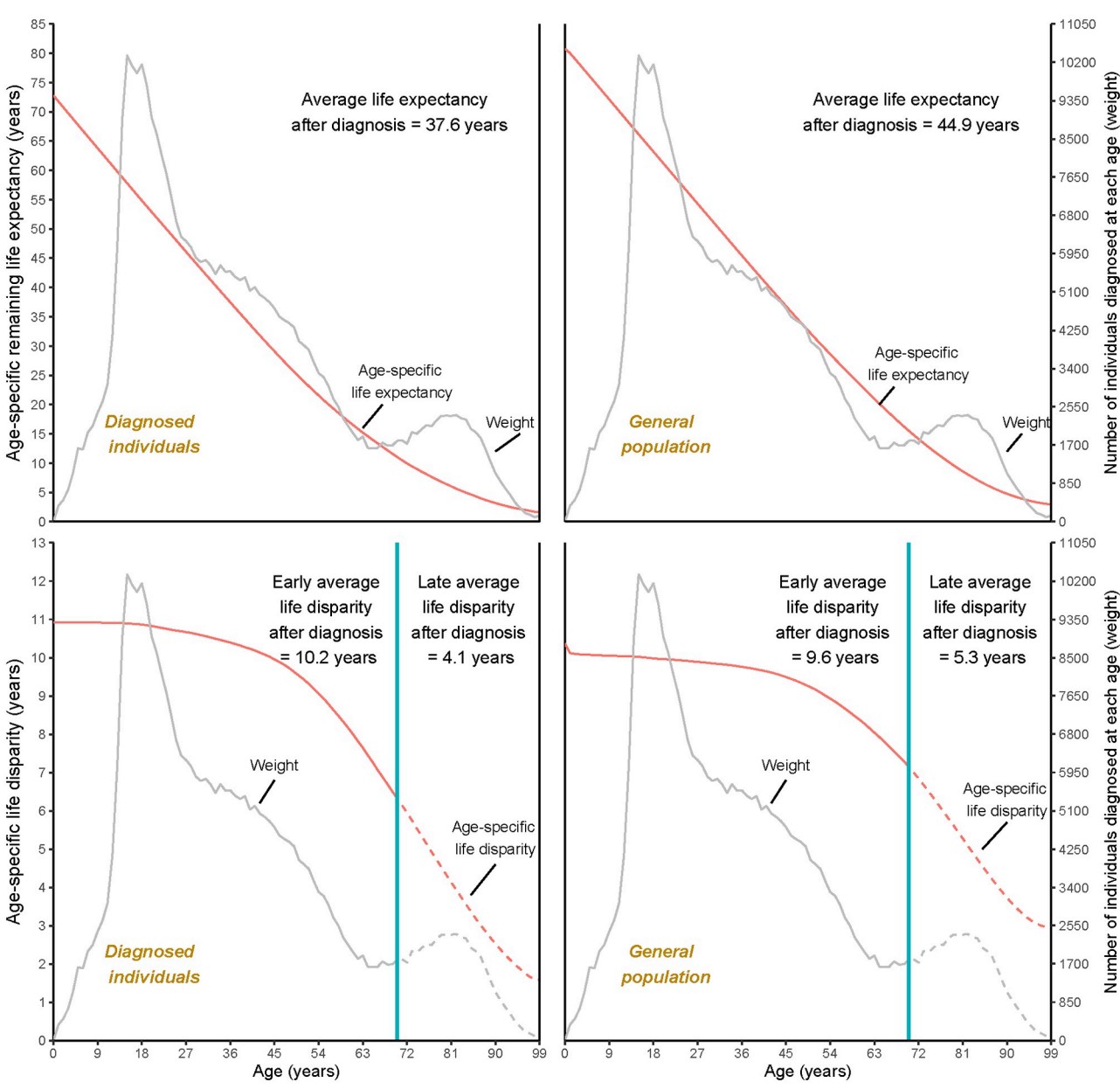

**Fig 1. Age-specific and average remaining life expectancy and life disparity (early and late) for women with mental disorders and women from the general population in Denmark.** Blue lines for the bottom figures represent the threshold age for women with any mental disorder used to separate the early average life disparity and late average life disparity for the two populations. The weights used to estimate average life expectancy and life disparity were the same for the two groups, namely, the number of diagnosed cases at each age.

Additionally, the figure shows the average metrics after disease diagnosis taking into account the number of cases at each specific age. The average remaining life expectancy after disease diagnosis for women with any mental disorders was 37.6 years, 7.3 years shorter than the general population (44.9 years), while their average life disparity after disease diagnosis was 9.5 years, 0.4 years greater than women from the general population (9.1 years). After considering the threshold age, the early average life disparity for women with any mental disorder was 0.6 years larger than women in the general population (10.2 years vs. 9.6 years), while the late

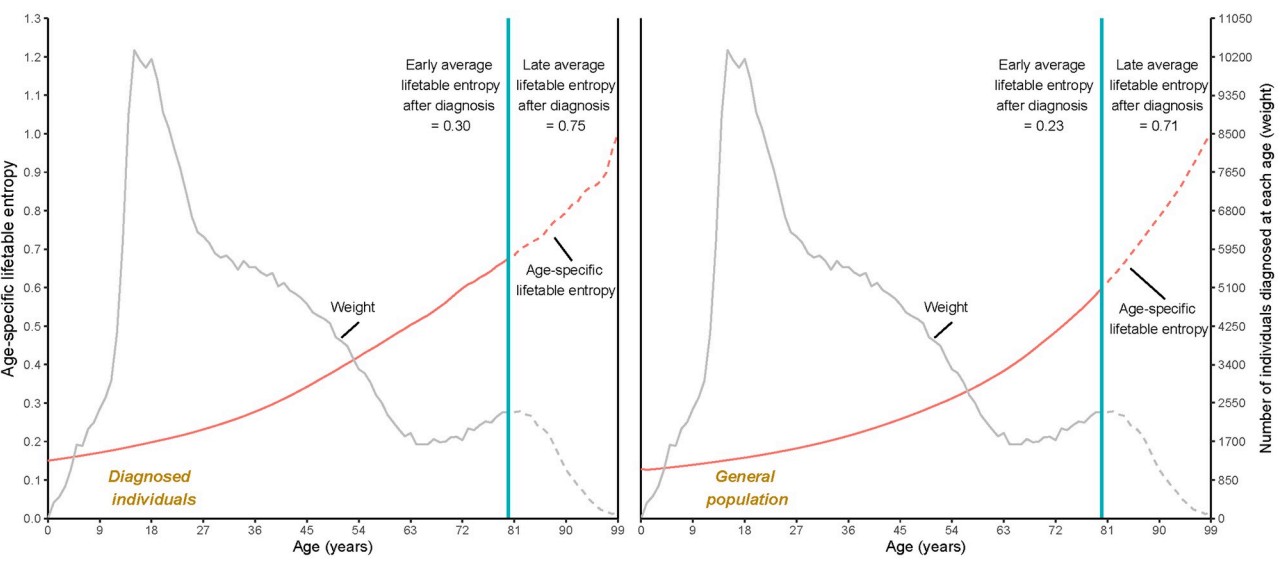

**Fig 2. Age-specific and average lifetable entropy (early and late) for women with mental disorders and women from the general population in Denmark.** Blue lines for the bottom figures represent the threshold age for women with any mental disorder used to separate the early average life entropy and late average life entropy for the two populations. The weights used to estimate average life entropy were the same for the two groups, namely, the number of diagnosed cases at each age.

average life disparity for women with a diagnosis was 1.2 years smaller than that for the general population (4.1 years vs. 5.3 years).

Fig 2 shows the relative measure of the average variation in lifespans for the observed two populations. Without considering the threshold age, estimates of the average lifetable entropy for women diagnosed with mental disorders and those in the general population were 0.33 and 0.27, respectively, which were consistent with the average life disparity after diagnosis. However, unlike the average life disparity, for both early and late average lifetable entropy, women with mental disorders experienced larger variation estimates than those from the general population. Specifically, for women with a diagnosis, the early and late average lifetable entropy were 0.30 and 0.75, while for women from the general population, corresponding estimates were 0.23 and 0.71.

## Discussion

In this study, alongside the average remaining life expectancy, we propose new mortality-related metrics to investigate the lifespan variation among people with particular diseases or conditions: the average life disparity and average lifetable entropy after diagnosis. By using data on women with mental disorders from Danish registries, our study replicated previous studies showing that women with a mental disorder had shorter average remaining life expectancy after diagnosis than women from the general population [12, 15]. More importantly, our results revealed that women with a diagnosis of a mental disorder also suffered from larger variation in lifespan compared with the general population.

The use of the average life disparity and average lifetable entropy after diagnosis provides a new perspective into the assessment of people's health associated with a given disease or condition. Within the field of epidemiology, apart from a range of mortality metrics (e.g., mortality rate ratio), life expectancy has been commonly used to evaluate the effect of particular diseases or conditions on premature mortality. However, the variation in lifespan has not received

attention from epidemiologists. By introducing this new metric, we estimate the heterogeneity in individuals' life years among those with a diagnosis, which would be concealed if only reporting average measures like life expectancy. Thus, we quantify the health gap for individuals with a given disorder from a new point of view. Practically, it draws attention towards the health inequality among those having diseases or conditions and concerns regarding their personal lifetime arrangements. At the macro level, it also calls for the necessity for the distribution of healthcare resources among the diagnosed people. Moreover, this new metric is an extension of the current advances in methodology in epidemiological research [4, 11, 12] as it also incorporates the observed age-at-onset, thus avoiding the assumption of a fixed age of onset of time-varying diseases and conditions, and can be applied to other major diseases. Additionally, individuals are followed from diagnosis, which means the estimates are conditioned on survival to that given age; thus, our method avoids any type of immortal time bias. This methodology guarantees a more accurate estimation of the effect of particular time-varying diseases that are not present at birth.

Considering the existence of the threshold age, we separated the average life disparity and average lifetable entropy after diagnosis into early and late components. Our results found that, compared with women from the general population, women diagnosed with mental disorders had a larger early average life disparity but a smaller late average life disparity. This is partly attributed to higher mortality rates for diagnosed individuals throughout their entire life, which would increase their life disparity at early ages but reduce their life disparity at late ages. Regarding the relative measure of variation in the length of lifespan, women diagnosed showed larger early as well as late average lifetable entropy than the general population. As lifetable entropy reflects the relative variation in the length of life compared to life expectancy at birth [32], it may present contrasting dynamics compared with life disparity [33]. For instance, higher mortality rates at old ages among diagnosed individuals would reduce the life disparity; however, as it would also decrease the corresponding life expectancy, it may increase the lifetable entropy (i.e., relative variation). This divergence reveals the disagreement when using different metrics to measure the variation in lifespan, which again underlines the crucial influence of the age-at-death distribution. This issue will be particularly important when investigating these metrics across periods as mortality/morbidity compression or expansion, particularly the mortality shift at older ages, would play an important role in affecting the longitudinal trend in lifespan variation [34, 35], which will also be applied to the average lifespan variation measures.

There are a few considerations to take into account. First, the proposed measures are based on mortality rates, which might not be accurate at very old ages (e.g., having extreme values) in some countries, particularly developing countries, which means that smoothing techniques for adjusting the observed mortality rates might be necessary. However, the ages in which there are more cases of mental disorders will have a higher weight on the average measures we propose; thus, extreme values might have a lower impact here than on standard non-averaged mortality metrics. Second, apart from the changes in the age-specific mortality rates, the varying distribution of the diagnosed cases at each age of onset will also contribute to the change in average lifespan variation measures. Thus, the reported shift in ages-of-onset [36] might have an impact on lifespan variation as well. Third, mortality shifts could potentially lead to false (de)compression effects as life expectancy is measured at fixed ages [37]. Thus, in the scenario in which those with a given disease experience reduced life expectancies, the differences in lifespan variation would be expected to be larger than the observed ones. Fourth, because the aim of this paper was to discuss metrics to estimate lifespan variation among those with a given disease, we have not calculated a measure of uncertainty. However, standard errors and confidence intervals may be obtained via non-parametric bootstrap.

In conclusion, this new mortality-related metric answers a new question in epidemiological research. Instead of merely focusing on average longevity, we drew attention to another dimension of mortality and analyzed the heterogeneity in lifespan among people diagnosed with a particular disease or condition. Moreover, we overcame the limitation that assumed a fixed age of onset of disorders or conditions and considered the underlying age-of-onset distribution. While this would not be necessary for conditions that are present at birth (or at a fixed age), it is important to take the age of onset into consideration when analyzing the impact of time-varying disorders or conditions in future epidemiological studies. This new health metric allows us to evaluate the influence of diseases or conditions on people's health more accurately and comprehensively, thus also providing important implications for the future design of public health policy and the distribution of healthcare resources.

## Supporting information

**S1 Table. Life table for women diagnosed with mental disorders after age 15 years based on observed mortality rates.**
(PDF)

**S2 Table. Life table for women from the general population.**
(PDF)

**S1 File. R codes.**
(PDF)

## Author Contributions

**Conceptualization:** Oleguer Plana-Ripoll.

**Formal analysis:** Yan Zheng.

**Funding acquisition:** Oleguer Plana-Ripoll.

**Methodology:** Yan Zheng.

**Supervision:** Oleguer Plana-Ripoll.

**Writing – original draft:** Yan Zheng, Oleguer Plana-Ripoll.

**Writing – review & editing:** Yan Zheng, Iñaki Permanyer, Vladimir Canudas-Romo, José Manuel Aburto, Andrea Nigri, Oleguer Plana-Ripoll.

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
