## [Decision Letter · Decision Letter 0]

3 May 2023

PONE-D-23-06555Lifespan variation among people with a given disease or conditionPLOS ONE

Dear Dr. Plana-Ripoll,

Thank you for submitting your manuscript to PLOS ONE. After careful consideration, we feel that it has merit but does not fully meet PLOS ONE’s publication criteria as it currently stands. Therefore, we invite you to submit a revised version of the manuscript that addresses the points raised during the review process. 

We look forward to receiving your revised manuscript.

Kind regards,

Binod Acharya

Academic Editor

PLOS ONE

Journal Requirements:

“This work was supported by a Lundbeck Foundation Fellowship to Oleguer Plana-Ripoll (R345-2020-1588). Oleguer Plana-Ripoll has also received funding from Independent Research Fund Denmark (grants 1030-00085B and 2066-00009B). The funders of the study had no role in developing the methodology, study design, data collection, data analysis, data interpretation, or writing of the report.”

Reviewers' comments:

Reviewer's Responses to Questions

**Comments to the Author**

1. Is the manuscript technically sound, and do the data support the conclusions?

Reviewer #1: Yes

Reviewer #2: Partly

Reviewer #3: Partly

2. Has the statistical analysis been performed appropriately and rigorously? 

Reviewer #1: Yes

Reviewer #2: Yes

Reviewer #3: No

3. Have the authors made all data underlying the findings in their manuscript fully available?

Reviewer #1: No

Reviewer #2: Yes

Reviewer #3: No

4. Is the manuscript presented in an intelligible fashion and written in standard English?

Reviewer #1: Yes

Reviewer #2: Yes

Reviewer #3: Yes

5. Review Comments to the Author

Reviewer #1: PONE-D-23-06555: Lifespan variation among people with a given disease or condition

This generally well-written article focuses on a method to estimate lifespan variation among people with a prevalent disease. While current demographic work has developed ways of measuring lifespan variation, combining these demographic ideas with the epidemiologic ones of disease prevalence has not been done before. In summary, this is a very well done combination of demographic and epidemiologic concepts. The methods seem robust, and the authors focus on a case study of Danish women with mental disorders. I have a few specific comments below:

Major comments:

1. Introduction: the introduction is well framed and outlines very well the justification for this study. A few comments: (a) “In the field of epidemiology, mortality metrics are fundamental for decision-making and resource distribution in the healthcare sector. “ The mission of epidemiology goes way beyond the healthcare sector, and I would argue that’s the sector where epidemiology hast the fewest to contribute (as opposed to social services planning, public health services planning, urban planning, etc.; in fact, line 70 mentions policymakers!); (b) I think a definition of life expectancy would be useful for a generalist journal such as plos one. This is especially important since the second paragraph mentions “a correct interpretation of the measure of life expectancy” but this interpretation is not provided.

2. Metrics, L123: the authors mention that limitations to prior methods include “using a fixed age of onset” but it isn’t clear to this reviewer (and maybe to other readers) what this limitation is about.

3. Metrics, average LE after disease diagnosis: if I am understanding equation 2 right, this would be similar to estimating an age-adjusted/standardized (direct) mortality rate, but where we would be weighting the residual ex by a fixed distribution of ages at onset (instead of by the actual distribution of ages at onset). Is this right? If it isn’t, it may be good to think how to reword this.

4. Metrics, early and late disparity: I suggest including an interpretation of e^dagger_0 and _1 in this section to help the reader understand how each one is interpreted differently.

5. Results, figures: the two figures are fantastic, and their labelling helps the reader. The only thing I’d add is an indication that the weight is fixed (footnote?)

6. Early vs late (results and discussion): if I am understanding things right, the authors find that Danish women with a diagnosed mental disorder live on average 7.3 fewer years than women without a diagnosis. They also find that life disparity is wider in general, but this is mostly driven by a wider life disparity before the threshold age vs a narrower one after the threshold age. However, for lifetable entropy women with a diagnosis have a wider variation both before and after the threshold age. The authors attribute this to higher overall mortality rates at any age for women with a diagnosis. Before the threshold age, this would increase lifespan variation (as higher mortalities before this age increase variation; see reference 21); however, after the threshold age, this would decrease lifespan variation. However, with entropy, as it takes into consideration ex, this phenomenon does not occur as it is more of a relative measure. Is this right? If it isn’t, it may be good to think how to reword this.

7. Immortal time bias: the authors don’t discuss this type of bias associated with diagnosis of conditions and I am wondering if they could think through and discuss its implications. My intuition is that since both ex and e^dagger_x are conditional on survival to a given age this wouldn’t be an issue. The same would apply to H_x since it just relies on ex and e^dagger_x, but I wanted to make sure the authors have considered this, as metrics “following a diagnosis” tend to be a classical example of immortal time bias.

Minor comments:

8. Abstract: the first sentence speaks about individuals, but I’d say that life expectancy is useful to compare longevity across populations, not so much individuals. In the second sentence, the authors say that “The variation in lifespan […] has not been estimated”, but I think it’s missing something about the variation in lifespan IN people with a given condition. Last, the sentence “Unlike the average life disparity, for both early and late average lifetable entropy, women with mental disorders experienced larger estimates than those from the general population. “ may benefit from some rewording to clarify (estimates of what?). For example “Unlike the average life disparity, both early and late average lifetable entropy was higher for women with mental disorders compared to the general population”.

9. Data: I don’t see any description of how the Danish VRS is linked with the Danish Psychiatric Central Research Register.

10. Metrics, L118: This sentence is missing something at the end (“among those diagnosed” with…?).

Reviewer #2: On lifespan variation among people with a given disease or condition

The paper presents some very interesting measures of life expectancy variation among people with a given disease, averaged over age at diagnosis. As such, the idea can be welcomed as it potentially provides a new way of looking at inequalities between people with a given diagnosis. Implicitly, inequality in life expectancy also suggests how quality of life may differ between the healthy and the sick: do the sick suffer both a shorter life expectancy and a longer period of severe disability before death, or at least could the period of severe disability before death be assumed not to be protracted? In fact, the rich discussion on the topic of mortality and morbidity compression/expansion started (with [1]) with this very important question in mind: if we extend life expectancy, do we also extend the period of severe disability before death? Apart from the individual quality of life before death, the question is also of public health concern: prolonged periods of disability imply higher health care costs as life expectancy increases. The answer to this question at the population level is negative: life expectancy increases, but disability and mortality are compressed into shorter periods of cohort life. There is value in extending this discussion to disease-specific contexts.

However, this brings me to the major problem of the paper, which should be corrected before publication: the authors do not relate their metrics to the traditional issues of mortality/morbidity compression/expansion. One, perhaps more intuitive, way of doing this might be (in addition to a refined comparison and discussion of the variation in life expectancy between people with different health conditions) to look at the temporal change in the metrics produced.

On a more technical side:

• Mortality shifts (in either direction) can lead to false (de)compression effects if life expectancy change metrics are calculated for fixed ages [2] - this seems to me to be an issue for the age-averaged metrics proposed in the paper as well. Consider a scenario where the mortality of those with a disease is shifted to younger ages compared to the main population. How would your metrics react (ideally, there should be no change in life expectancy variation)?

To illustrate, I run a simple scenario that mimics your parameters using the 2014 Australian life table: I compare the original life table with one where the mortality curve is shifted 7 years younger along the age axis. Life expectancy at age 39 is closest to your average life expectancy of 44.9 in the general population; it is 7 years shorter (because of the shift) in the worse mortality case. Looking at e-dagger at 39, one would find that it shrinks by 0.3 years from 9.2 in the original to 8.9 in the 'shifted' scenario. Thus, a pure shift induces a false compression for those 'diagnosed' at age 39. This spurious effect is opposite in sign to the one you report, but comparable in magnitude! Thus, after somehow correcting for the shifting effect, one would get an effect almost twice as large as the one reported in the paper. /For entropy, the effect of the shift is more complex.

• Entropy is a different story from e_dagger. Unlike e_dagger, which more explicitly reflects variation in age at death, H is useful as a tool to roughly infer the effects of proportional mortality reduction/increase on life expectancy [3]. This is an interesting research perspective in itself, perhaps even more so in the context of reducing cause-(diagnosis-) specific mortality. I suggest you address this aspect in your paper to better link it to the existing literature on H.

After addressing the points above, your discussion section may also need to be redesigned.

Overall, the idea is great indeed, but the presentation needs a major improvement in order to fully realize it (the idea).

Possible typos:

• In eq. (1), shouldn’t you use another index (say i for individuals), not age x?

• Eqs (4), (5) – shouldn’t they have different denominators n0 and n1, not the total n?

Refs:

1. Fries J.F. Aging, natural death, and the compression of morbidity // N. Engl. J. Med. 1980. Vol. 303. P. 130–136.

2. Ediev D.M. Decompression of Period Old-Age Mortality: When Adjusted for Bias, the Variance in the Ages at Death Shows Compression // Math. Popul. Stud. 2013. Vol. 20, № 3. P. 137–154.

3. Keyfitz N., Golini A. Mortality comparisons: The male-female ratio // Genus. 1975. Vol. 31, № 1/4. P. 1–34.

Reviewer #3: The manuscript proposes to estimate lifespan variability of individuals that have been diagnosed mental disorders. The motivation is that while it is common practice to estimate average lifespan (via life expectancy) of individuals with a specific disease, it can be useful to measure also inequality of lifespans. Then the new metric is applied to Danish register data.

the wya

The paper is overall well written, and the idea of using a measure of lifespan disparity is interesting. However I have one concern about the way this new metric is presented. If authors' intention is proposing a new measure, then I think they should devote more space to mathematical/statistical properties of if. What is the uncertainty related to this measure? How much is sensitive to outliers or to values at higher ages where death rates - and thus life expectancies, are more volatile? Some diagnostic and discussion of this measure should be provided.

I understand that this would mean a drastic transformation of the paper but at the present time, the manuscript lies in between a methodological paper and an applied one. I think this ambiguity should be fixed.

MINOR:

- Table 1 refers to a "cohort" of individuals. It is not entirely clear to me what do you mean by "cohort" here. To my knowledge a cohort is a set of individuals that underwent a specific event (the "origin" event) in the same year. It is clear that the starting event is a diagnosed mental disorder, but it is not clear what is the common year. Maybe you could explain a little more about this.

- Figures come at a very low quality. I understand that in some cases is the editorial manager to transform high quality images into blurred figures, however I recommend authors to ensure that figures have a good definition

- I applaud to authors' choice to share the R code, but at the same time I suggest to do that in a way that makes it more readable, for instance using comments to explain what a specific chunk is intended for

6. PLOS authors have the option to publish the peer review history of their article (what does this mean?). If published, this will include your full peer review and any attached files.

Reviewer #1: No

Reviewer #2: **Yes: **Dalkhat M. Ediev

Reviewer #3: No

---

## [Author Response · Author response to Decision Letter 0]

21 Jun 2023

Please see the rebuttal letter attached

---

## [Decision Letter · Decision Letter 1]

21 Aug 2023

Lifespan variation among people with a given disease or condition

PONE-D-23-06555R1

Dear Dr. Plana-Ripoll,

We’re pleased to inform you that your manuscript has been judged scientifically suitable for publication and will be formally accepted for publication once it meets all outstanding technical requirements.

Kind regards,

Binod Acharya

Academic Editor

PLOS ONE

Additional Editor Comments (optional):

Reviewers' comments:

Reviewer's Responses to Questions

**Comments to the Author**

1. If the authors have adequately addressed your comments raised in a previous round of review and you feel that this manuscript is now acceptable for publication, you may indicate that here to bypass the “Comments to the Author” section, enter your conflict of interest statement in the “Confidential to Editor” section, and submit your "Accept" recommendation.

Reviewer #1: All comments have been addressed

Reviewer #2: All comments have been addressed

Reviewer #3: All comments have been addressed

2. Is the manuscript technically sound, and do the data support the conclusions?

Reviewer #1: Yes

Reviewer #2: Yes

Reviewer #3: Yes

3. Has the statistical analysis been performed appropriately and rigorously? 

Reviewer #1: Yes

Reviewer #2: Yes

Reviewer #3: Yes

4. Have the authors made all data underlying the findings in their manuscript fully available?

Reviewer #1: No

Reviewer #2: Yes

Reviewer #3: Yes

5. Is the manuscript presented in an intelligible fashion and written in standard English?

Reviewer #1: Yes

Reviewer #2: Yes

Reviewer #3: Yes

6. Review Comments to the Author

Reviewer #1: The authors have satisfactorily addressed all of my comments, and I have no further concerns of this robust manuscript.

Reviewer #2: (No Response)

Reviewer #3: I thank the author for this new version of their manuscript and for their efforts in considering my comments in their revised paper. The latter is now, to my view, improved. I'm still convinced that such a new metric deserves a wider set of data to tested on, eventually using simulated data, thus I encourage authors to apply the proposed measure to other data.

The figures keep on appearing very blurred, but this is probably due to how editorial manager creates the final pdf, probably reducing the resolution of figures. Please check this with editorial staff.

7. PLOS authors have the option to publish the peer review history of their article (what does this mean?). If published, this will include your full peer review and any attached files.

Reviewer #1: No

Reviewer #2: No

Reviewer #3: No

---

## [Editor Report · Acceptance letter]

24 Aug 2023

PONE-D-23-06555R1 

Lifespan variation among people with a given disease or condition 

Dear Dr. Plana-Ripoll:

I'm pleased to inform you that your manuscript has been deemed suitable for publication in PLOS ONE. Congratulations! Your manuscript is now with our production department. 

Kind regards, 

on behalf of

Mr. Binod Acharya 

Academic Editor

PLOS ONE